# Emergent Social Capital during the Coronavirus Pandemic in the United States in Hispanics/Latinos

**DOI:** 10.3390/ijerph20085465

**Published:** 2023-04-11

**Authors:** Jennifer Contreras, Alexandra Fincannon, Tasneem Khambaty, Ester Villalonga-Olives

**Affiliations:** 1Department of Practice, Sciences, and Health Outcomes Research, University of Maryland School of Pharmacy, Baltimore, MD 21201, USA; 2Department of Psychology, University of Maryland, Baltimore County, Baltimore, MD 21250, USA

**Keywords:** bonding social capital, bridging social capital, trust, Latinos, COVID-19, misinformation

## Abstract

The coronavirus pandemic has drastically impacted many groups that have been socially and economically marginalized such as Hispanics/Latinos in the United States (U.S.). Our aim was to understand how bonding social capital, bridging social capital, and trust played a role in Hispanics/Latinos over the course of the COVID-19 outbreak, as well as explore the negative consequences of social capital. We performed focus group discussions via Zoom (n = 25) between January and December 2021 with Hispanics/Latinos from Baltimore, MD, Washington, DC, and New York City, NY. Our findings suggest that Hispanics/Latinos experienced bridging and bonding social capital. Of particular interest was how social capital permeated the Hispanic/Latino community’s socioeconomic challenges during the pandemic. The focus groups revealed the importance of trust and its role in vaccine hesitancy. Additionally, the focus groups discussed the dark side of social capital including caregiving burden and spread of misinformation. We also identified the emergent theme of racism. Future public health interventions should invest in social capital, especially for groups that have been historically marginalized or made vulnerable, and consider the promotion of bonding and bridging social capital and trust. When prospective disasters occur, public health interventions should support vulnerable populations that are overwhelmed with caregiving burden and are susceptible to misinformation.

## 1. Introduction

The coronavirus pandemic revealed several vulnerabilities in the United States (U.S.), especially longstanding health disparities in people from racial and ethnic minority groups [1,2,3]. According to the CDC, Hispanics/Latinos have been the most affected by the COVID-19 pandemic, demonstrating disproportionally high rates of infection and death [4]. The high rates have been attributed to the substantial racial and ethnic disparities that exist in the American health care system and the prevalence of comorbidities, which are exacerbated by limited access to care [4,5]. A cross-sectional study by Lee et al. found that racial and ethnic minorities reported significantly higher perceived provider discrimination and poorer health compared with their white counterparts. The authors suggest that when individuals experience provider discrimination, they may become dissatisfied with the health care system and reduce health care utilization [6]. At the population level, lower health care utilization involves increased negative health outcomes in groups with higher risk of COVID-19 vulnerability, such as Hispanics/Latinos [6].

Social capital encompasses resources such as trust, social cohesion, and information channels within social networks [7,8,9,10]. It is a social determinant of health that influences health outcomes [11,12,13]. Strong social networks have resulted in important health-related benefits in the context of prenatal, asthma, and mental health care, among others [14,15,16]. However, the relationship between social capital and health often depends on the quality and type of relationships among members of a given community. Bonding social capital helps to provide social support, financial assistance, and solidarity to individuals within close-knit relationships. Bridging social capital helps to build trust and maintain channels of communication among individuals from different social groups and across cultures with dissimilar characteristics [17]. In short, social ties, whether in-group (bonding) or between groups (bridging), provide support, assistance, and collective action [18]; both bridging and bonding social capital can provide benefits to Hispanics/Latinos and contribute to reducing health disparities. In the literature, two case studies demonstrated how social capital was essential for Hispanics/Latinos living in low-income neighborhoods to acquire resources for disaster recovery during catastrophic events such as natural disasters [19,20,21]. Additionally, there is evidence that trust played a positive role in disaster preparedness during Hurricane Katrina, which impacted many Hispanics/Latinos in the U.S. Gulf Coast [19]. According to Fukuyama, honesty and cooperative behavior within a community are the products of trust [22]. Additionally, in his book, *Trust: Social Virtues and the Creation of Prosperity*, Fukuyama highlights how trust is a part of social capital and can be observed in either small social groups, such as families, or large groups, such as countries. Trust has had important implications for adherence to CDC recommendations for preventive measures such as wearing masks, social distancing, and getting vaccinated for COVID-19 [23,24]. Previous studies have shown that when government agencies disseminate health-related information through trusted entities, the acceptability of the information increases [25,26].

Although it is important to recognize the evidence that social capital is associated with positive health outcomes, there is evidence that social capital is also associated with negative health outcomes [27]. Strong bonding social capital in Hispanics/Latinos often results in dense social connections within the group with information, emotional support, and solidarity, all of which are beneficial to the Hispanic/Latino population. This phenomenon could result in positive outcomes such as encouragement to implement healthy lifestyle changes [28]. However, social capital may also result in isolation from interactions with individuals outside of their close-knit social circles [17]. Higher levels of bonding social capital may remove a community’s ability to commiserate and build connections with other groups, which consequently supplants bridging social capital and leads to making vertical connections less likely [29]. In one sense, there is a zero-sum game with social resources, with communities typically unable to have high levels of multiple types of social capital [30]. Additionally, when relationships have high levels of trust and social cohesion, they may result in adverse outcomes, such as sharing misinformation [31]. For example, in social groups with elevated levels of trust and social cohesion, there is the potential for individuals in the group to hear misleading information and accept it as fact [27]. Consequently, this can make it difficult for clinical providers caring for patients that belong to these social groups, regarding addressing misinformation and encouraging preventive measures and vaccination.

Previous studies broadly demonstrated how bonding and bridging social capital can result in both positive and negative health outcomes in response to the COVID-19 pandemic in the general U.S. population [32]. However, to the best of our knowledge, social capital has not been studied in the U.S. Hispanic/Latino community during the pandemic, although previous studies have examined how social capital was beneficial to low-income Hispanics/Latinos living in the U.S. during disruptive events such as Hurricane Katrina and Hurricane María [19,20,21]. What remains unknown, however, are the manifestations of social capital in U.S. Hispanics/Latinos during the COVID-19 pandemic. We believe that the impact of social capital on the Hispanic/Latino population during a pandemic can inform public health and healthcare workers in developing interventions for emergency responses and recovery when other disasters happen. Therefore, our qualitative study aimed to examine how bonding social capital, bridging social capital, and trust played a role in the Hispanic/Latino community over the course of the COVID-19 outbreak, as well as explore the negative consequences of social capital. This paper will describe the process through which social capital shaped the responses of Hispanics/Latinos during the pandemic in the U.S.

## 2. Materials and Methods

### 2.1. Study Design and Setting

We conducted a qualitative study using focus groups. The focus groups were conducted in Spanish via Zoom between January and December 2021 and comprised of eight focus groups (Total n = 25 participants) with Hispanics/Latinos residing in Maryland; Washington, DC; and New York. We employed focus groups to facilitate dynamic conversations between study participants. The sample size was acceptable according to other qualitative studies and managed to achieve saturation for the topics included in the semi-structured discussion guide [33,34,35,36]. As Chamlee-Wright et al. describe in their work, qualitative approaches clarify questions about how people react to unprecedented situations in terms of emergent forms of social capital, dark sides of social capital, and trust [37]. Furthermore, the ways in which individuals build their narratives are difficult to capture through surveys and quantitative analysis. The semi-structured focus group discussion format creates a window through which researchers can see different forms of social capital more clearly. In this case, a qualitative, semi-structured format was an effective way to access information from community members and frame their experiences during the pandemic. The focus group discussions were administered with a semi-structured discussion guide by a trained interviewer that had questions on sources of support, vulnerabilities, and trusted forms of information and government/public health officials. The semi-structured discussion guide was based on previous qualitative work on social capital [18] and is included in Appendix A. We report that our study followed the COREQ (Consolidated Criteria for Reporting Qualitative Research) guidelines [38].

### 2.2. Research Team and Reflexivity

The focus groups were facilitated by the senior author and an interviewer contracted from Westat, a research services firm. The senior author and the interviewer are both female, fluent in Spanish, self-identify as Hispanic or Latino, and trained in qualitative research methods. Additionally, the senior author is a social epidemiologist with expertise in the social determinants of health within the Hispanic/Latino community.

### 2.3. Participants (Sampling)

Purposive sampling captured a diverse sample in terms of employment, so a range of experiences could be captured in the focus groups, from essential to non-essential workers. The goal of our sample recruitment was to have focus group discussions with potential participants after the stricter lockdowns had lifted and the vaccine was readily available in 2021 to understand the range of experiences that participants had at the beginning of the pandemic through a current point in time.

To be eligible for this study, individuals had to (a) be adults aged ≥ 18, (b) self-identify as Hispanic/Latino, and (c) be fluent in Spanish. Participants were recruited via Esperanza Center and Heritage Care, Inc. in Maryland and Mount Sinai Health System Center (MSHS) of Spirituality and Health in New York. We selected Maryland and New York to recruit participants because the study investigators had existing community partners in these locations to assist with recruitment. The staff at Esperanza Center, Heritage Care, and MSHS recruited a total of 55 participants. Of those, 33 accepted to participate initially, and 8 were not available at the time of the focus group discussions. Thus, the focus groups had a total of 25 participants.

### 2.4. Data Collection

Participants were asked to complete an online survey via Qualtrics to collect sociodemographic data (age, sex, education, income, and employment). All focus group conversations were audio recorded and transcribed. We conducted focus groups until thematic data saturation, defined as when a thorough understanding of the perspectives of the participants was obtained with few or no new concepts emerging in subsequent data collection.

### 2.5. Analysis

Transcripts from the focus groups were analyzed in Spanish using Atlas TI, a computer-assisted qualitative data analysis software. Coding and analyses were performed according to the principles of grounded theory and thematic analysis [39,40]. These approaches were used because they are widely accepted practices in qualitative research and reflect the experiences that were shared by the participants. Both grounded theory and thematic analysis used simultaneously can define and redefine themes to fit the data [41]. During analysis, one coder fluent in Spanish derived themes from the study data. Quotes were translated to English by the first author of this manuscript and reviewed by the senior author and an external reviewer to ensure concepts were accurate in English. Participant quotes highlighting key findings are presented in the results. Participants did not provide any feedback on the findings.

## 3. Results

Our sample was mostly adults between the ages of 25 and 30 (26.1%) and females (76.2%) (Table 1). We had an equal representation of essential workers (42.9%) and non-essential workers (42.9%). A total of 11 individuals from the sample reported earning less than $30,000 (45.5%), and 5 reported earning more than $100,000 (18.2%). About half of our sample reported not having a college degree, used Spanish as their primary language (71.4%), and were insured (66.7%).

The thematic analysis findings are presented in Table 2. Four major themes and fourteen subthemes were identified in the data: (a) bonding social capital (subtheme(s): solidarity, virtual meetings, and isolation); (b) bridging social capital (subtheme(s): information exchange); (c) trust (subtheme(s): vaccine distrust, mistrust in politicians, mistrust in scientists, mistrust in medicine, trust in the vaccine, trust in politicians, trust in scientists, and faith); and (d) dark side of social capital (caregiving burden and misinformation). Themes and subthemes were consistent throughout all focus groups. In the reporting of our results, we give examples of quotes with the main themes and subthemes for each quote. Finally, we report emerging themes identified in our analysis, which are related to topics not included in the focus group discussion guide. Emergent themes were endorsed by some participants only and not the majority.

### 3.1. Bonding Social Capital

We found that participants gained solidarity from their neighbors and individuals in their community. One participant remarked that during the pandemic, residents in his apartment building held a meeting for the first time in his recollection (Table 2). Another participant shared how he received support from his neighbors when he and his family were infected with COVID-19.

“*Something that I noticed more of was solidarity. For instance, I now know almost all my neighbors in the building. There was a moment when the building didn’t have water, and in response, my neighbors gathered to have a meeting, which is something that never happened before.*”(Male, Hospital Worker)

“*I think that people have gained more support from friends or neighbors. Because my wife and I were infected with COVID, and honestly our neighbors were very supportive. They brought us something to eat, or sometimes tea to drink. I think that I gained a little more trust with my neighbors.*”(Male, occupation not ascertained)

Two other participants shared how video calls helped families and friends remain connected during the shut-down/quarantine period. For instance, one participant that works at a hospital as an essential worker noticed how families remained connected despite the strict measures that hospitals implemented to prevent COVID-19 infections.

“*I think that family unity is important, because even if it were by phone or by a FaceTime call, families remained connected. They would ask me for a session to be held so that various members who were in different homes could see the patient. In our culture, there is a union and a family love that is different from other cultures.*”(Female, Hospital Worker)

However, one participant shared how living in New York when it was the epicenter of the pandemic made them feel lonely and isolated. They expressed that video calls via Zoom could only do so much to relieve their feelings of loneliness; the only remedy for their feelings of isolation was visiting family. Another participant shared a similar experience and discussed how virtual meetings cannot compare to the quality of face-to-face conversations with friends and peers. All participants agreed that despite being able to connect with family and friends via Zoom, the platform could not entirely relieve their feelings of isolation.

“*The beginning of the pandemic was very hard for me, because I was living in New York completely alone and I had no family nearby. Before, I had many friends and co-workers that helped me a lot when I first moved to New York. But with the pandemic, everyone was separated. As a result, I could not see my friends, and the interaction I had with my colleagues was simply over Zoom. The first few months of the pandemic were hard, but after I went to visit family, I began to feel better.*”(Female, Research Coordinator)

“*I socialized with an LGTBQ group from a Catholic parish, which is one of the few in Manhattan, but it has been very difficult for me to socialize with them during the pandemic because we only socialized through social media and video calls, which does not compensate for hugs. It also doesn’t compensate for close conversations, or having the person in front of you, for you to see how that person really is doing.*”(Male, Court Translator)

The lack of in-person interactions materialized as feelings of isolation. Bonding social capital was helpful to relieve those feelings. However, for most participants, especially those in New York City, the manifestations of bonding social capital were not enough.

### 3.2. Bridging Social Capital

There were several demonstrations of bridging social capital in the focus groups. Participants shared how they provided vital information to individuals in need that otherwise may not have had knowledge of available resources. One participant who is a doctor working for a community health organization shared how he used his profession to help other Hispanics/Latinos. He provided accurate information on COVID-19 prevention and addressed coronavirus misinformation and vaccine myths that may have been preventing others from getting vaccinated.

“*As a medical professional, my job allows me to help and guide people, but I can’t prescribe or administer medications because of limitations from my license. However, I can focus on the preventive part of medicine, which includes demolishing cultural myths that prevent people from getting vaccinated right now.*”(Male, Internal Medicine Doctor, Community Health Organization Worker)

Another participant shared how her interactions with the home health workers caring for her mother allowed the participant the opportunity to educate the workers on the vaccine and alleviate their apprehension about receiving it. This participant is a semi-retired social worker and community worker with knowledge of COVID-19 vaccination resources available to the caregivers.

“*My mom receives services from a retirement home and most of those workers are Latinas, from the Caribbean. Initially, I asked them, ‘what information is your company giving you regarding COVID, or regarding the vaccine?’ They told me that they were not receiving information and their personal notions were out of fear. I had to take the initiative to try to refer them to places in Spanish that informed them about COVID, and gave them information verbally, and gradually they changed their minds about the vaccine. Now one of them is fully vaccinated and the other one I helped her get vaccinated.*”(Female, Semi-Retired Social Worker)

One participant recounted how she was able to learn about the vaccine because she was an avid volunteer at her local hospital and attended lectures given by the doctors there.

“*One of the very famous doctors we have at Mount Sinai taught a lecture about everything that had to do with the vaccine. The lecture covered the materials and the proteins they used in the vaccine. We were informed in detail. Especially for many of the people who are already older, who did not want to get the vaccine, he explained it very well and in detail everything about the vaccine, and how you are going to help each one, where to go. He gave us a lot of information.*”(Female, Hospital Spiritual Care Volunteer)

Lastly, a participant noted how he was able to remedy the lack of accessible information in his community through an outreach program that allowed him to educate community members and register them for the vaccine over the phone.

“*We call the person, and we ask if they’ve been vaccinated, and if not, we complete the registration … that is why we go above and beyond, to improve our access to them … this is a direct action that we can do and that can be multiplied. In fact, I think that this program as a health promoter, has been for them a solution to many of their problems, or at least try to solve these problems and try to show them that it can be accessed, that with the necessary tools that we can provide, they can reach the systems, since the system is difficult.*”(Male, Internal Medicine Doctor, Community Health Organization Worker)

Manifestations of bridging social capital demonstrate how participants with access to reliable information about community resources were able to connect people in need of services that were unaware of the assistance available. Additionally, these knowledgeable individuals were able to advocate for the vaccine and increase inoculation via their peers. Vaccine advocates had higher levels of educational attainment and experience in health and human services.

### 3.3. Trust

In all focus groups, we observed a clear divide between participants when asked about trust, e.g., whether they trusted the vaccine or scientists. Reasons for mistrust included noticing vaccinated individuals still getting infected with COVID-19 and the expedited development time for the COVID-19 vaccine. Most participants also discussed their mistrust of politicians, with one person saying they trusted neither the government nor scientists because of the lack of transparency from both. Another interesting topic was medical mistrust and its role in hindering the Hispanic/Latino population from seeking medical services.

“*I have decided to hold off getting vaccinated because I have seen cases of people who did not have the virus and then got it by getting vaccinated, or because I also heard in the news that certain people that got vaccinated died two hours later. This creates a bit of fear because you do not know what reactions you are going to have in your own body.*”(Female, Essential Daycare Worker)

“*The only objection I have about the vaccine is the short amount of time it was developed. All medications have some side effect, but in the case of the vaccine we still don’t know much, and it may take up to 10 years for us to know. That is the only opposition that I have regarding the vaccine.*”(Female, Homemaker)

“*I don’t trust any politician. I feel like a lot of things are being done just to ease people’s fear, including vaccinations. They are rushing vaccinations without having long trials to see if it really is good. I don’t trust the scientists, or the government, in how they are handling the pandemic.*”(Female, Essential Daycare Worker)

“*There is a lot of mistrust among the Latino population with medical services. I remember that many patients have told me of individuals that they know that go to the hospital that arrived alive and have not left.*”(Female, Essential Hospital Worker)

On the other side of the divide, some participants reported trusting the vaccine and scientists, and a few said that they trusted politicians. One offered this perspective: because politicians had to share leadership with scientists that knew how to handle the pandemic, politicians benefited from trust in scientists and science. Participants also shared their trust in God and how their faith helped them cope with the uncertainty surrounding the vaccine.

“*I’m not afraid of vaccines. It has been almost 30 years that I do not know what a cold is, because I have a habit between October and November to get vaccinated for influenza. I know they are efficient and that’s why I get vaccinated with everything they offer me in my medical plan.*”(Male, Retired, Part-Time Construction Worker)

“*The difference with this crisis is that this is a health crisis where a politician must share the leadership space with the scientific and medical community. Right now, Dr. Fauci, even if he doesn’t want to, he has become an example, because he is the expert, and he must come out and show his face. He explains to the public through social media or television or whatever. The politician will always be a politician with or without a virus. I believe that in this case when the stage or scene is shared with the scientist, the situation is already different, and I am confident.*”(Male, Court Translator)

“*At first, I said that I was not going to get vaccinated, because some people said that it was going to change your DNA. I don’t like listening to people, and I always believe a lot in my God.*”(Female, Hospital Spiritual Care Volunteer)

### 3.4. Dark Sides of Social Capital

The focus groups revealed demanding relationships due to the caregiving burden and the spread of misinformation among bonded relationships. Many participants reported that they were caregivers for the most vulnerable members of their family, such as parents or grandparents. One participant shared how he would call his grandmother to check in on her periodically because he was worried about her being alone during the pandemic; he eventually went to stay with her since he was teleworking at the time. Another participant mentioned that it was difficult to take care of their mother with dementia.

“*When I was in New York, I spent most of the pandemic there. I was always calling my grandmother, checking-in to see that she was fine. As [the pandemic] was going on, I began to worry much more about my grandmother. Mostly about her being alone. I decided to come and stay with her, since I am still teleworking. At that time, I stopped worrying about myself, but worried more about her well-being. I didn’t want her to get out of the house and get infected. Now, I feel better being here with her. I feel more at peace and happy that she already has someone, and that she is not alone.*”(Female, Research Coordinator)

“*Now that we are all in the house, it has not been easy. My mother obviously with dementia has many needs and a lot of confusion. Although she has services, they are partial, so I frequently must talk to her, and reassure her everything will be okay.*”(Female, Semi-Retired Social Worker)

While some participants caring for older family members received some benefit from knowing their loved ones were doing well in their care, it is important to recognize that others experienced the burden of caregiving. Furthermore, as the pandemic progressed, a growing ancillary concern was the amount of misinformation that was being spread among close-knit relationships. A participant that is a hospital volunteer revealed that they had received misleading information from one of their friends.

“*As much as one always tries to convince Hispanic people to go to the doctor so that they can take better care of themselves, they just don’t go, and they mostly rely on home remedies. For example, a friend of mine told me, “Don’t get the vaccine. I’ll bring you some pills from Santo Domingo instead that won’t give you COVID.’ However, I don’t believe in any of that.*”(Female, Hospital Spiritual Care Volunteer)

### 3.5. Emergent Theme: Racism and Discrimination

Participants shared their observations of racism and discrimination in the Hispanic/Latino community. The lack of resources available to Hispanics/Latinos was discussed, especially for those that do not have legal status in the country. Participants listed the lack of safety-net resources, such as access to the internet or a computer, health insurance, and financial assistance. Without resources and assistance to connect to essential healthcare services, many Hispanics/Latinos delay visits to the doctor or do not use medical services, which can result in negative health outcomes. Many Hispanics/Latinos are underinsured, which prevents them from benefitting from the usual sources of care out of concern for out-of-pocket costs [42].

“*Some Latinos do not have safety-net resources or social security that the rest of the population that has legal status have. Most of our population do not have a legal status in this country and do not have access to many things.*”(Female, Hospital Worker)

“*We have seen that the systems have been designed to have access to vaccine information or appointments, for many members of our community, particularly the elderly, but it requires a computer, which they do not have access to, or do not have access to the Internet. If people are working, they must make a phone call, but the wait is so long that they can’t be on the phone waiting for an appointment.*”(Female, Semi-Retired Social Worker)

“*I work with people who are undocumented, and if they feel sick, they don’t have the money to pay for medical consultations and they’re afraid to go to the clinics. I have worked with many clients who have had COVID, and they are very vulnerable because they either don’t have the academic knowledge or have the education, and they do not speak English well. They also do not have an insurance plan or if they do, they are people who need so much, so they go to work even when they are sick.*”(Male, Court Translator)

“*Access to health services and health insurance is obviously a very serious problem in our community. All people who work and do not have medical insurance, they often do not seek medical services. The issue of access is also that the medicines are very expensive, and even though you have insurance, you must pay a co-pay or whatever. Sometimes you must make the decision of either paying the rent and the food for the month or pay for the medicines.*”(Female, Semi-Retired Social Worker)

## 4. Discussion

Four key findings resulted from our qualitative study of social capital in the Hispanic/Latino community during the pandemic. First, we found that bonding social capital provided benefits to focus group participants. For example, many described new-found solidarity with others while coping with COVID-19-related challenges. Interestingly, many participants stated that video calls helped them remain connected with family and friends but did not relieve feelings of isolation. Second, we identified several examples of bridging social capital, which can be used as a tool to disseminate accurate information on COVID-19 prevention and vaccination, thereby addressing misinformation. Third, we identified how trust in the government and public health professionals played an important role in influencing Hispanics/Latinos to adhere to pandemic preventive measures; this supports previous findings from similar studies [25,43,44]. Fourth, we observed the dark side of social capital, which encompasses the negative effects of social capital, such as excessive demands being placed on small groups of people, loneliness, and the exchange of misinformation [27]. In our study, many participants shared how they were caring for chronically ill family members and would often worry about transmitting coronavirus infection. Some participants also reported experiencing caregiving burden during this time, which supports prior findings [45]. Another example of the dark side of social capital that we found in our study is how easily misinformation can spread in close relationships, including how difficult it was to persuade others to refuse home remedies for coronavirus prevention. Acceptability of home remedies is prevalent in the Hispanic/Latino community, which makes it easier for misinformation to be spread and followed [46]. Importantly, Nooteboom observed that in instances where trust and social cohesion are very close, there is a risk of adverse effects, such as accepting myths as truth to prevent the termination of a valued relationship [31]. This negative consequence of social capital remains a barrier in clinical settings especially, where misinformation can have extremely bad consequences.

We observed the emergent theme of racism and discrimination, specifically, participants who mentioned institutional racism and discrimination in the healthcare system. In the U.S., structural racism is experienced by historically marginalized racial and ethnic groups, which often leaves them at a disadvantage compared with their White counterparts [47,48]. An example of racism and discrimination shared by participants was unequal access to safety-net services, which is consistent with the literature [49,50,51,52]. Based on focus group responses, migratory status appears to be one of the reasons many Hispanics/Latinos did not have medical insurance coverage, either private insurance or Medicare/Medicaid. Consequently, this leaves individuals without legal status excluded from safety-net resources available from health care institutions and other public social welfare organizations, making them especially vulnerable in the context of a global pandemic.

Our findings are consistent with other studies that have examined social capital during catastrophic events in the Hispanic/Latino community [19,20,21]. Similar to our findings, in a qualitative study by Roque et al. that examined the successful use of social capital in post-disaster responses during Hurricane María in Puerto Rico, it was observed that manifestations of bonding and bridging social capital in low-income neighborhoods enhanced resiliency in the communities [19,20,21]. In a qualitative study by Messias et al. that examined the dynamics of social networks among Latino survivors of Hurricane Katrina, the findings suggest that social capital helped with information gathering, decision-making, and acquiring resources. The Messias et al. study also found that post-recovery social networks were strained, and structural racism strained access to crucial information and resources, all of which were similar to our findings in examining social capital in the Hispanic/Latino community during the COVID-19 pandemic. However, our study additionally explores and provides information about the dark side of social capital in disruptive events in the Hispanic/Latino population, which has not been previously examined in other studies [19,20,21].

Our study had limitations, such as the unequal representation of men in our sample. However, we were able to obtain an equal representation of both essential and non-essential workers across all ages. Nevertheless, there is the potential for selection bias because our sample included a few individuals that had high levels of educational attainment and high incomes, which are not consistent with the target population overall. Another limitation of our study was that we did not follow up with participants after the focus group discussions; therefore, we cannot evaluate any evolution over time. Despite the limitations of our study, there were several strengths as well. For example, we completed focus group discussions with individuals from three large U.S. cities. In diversifying our focus groups, we were able to observe different experiences, especially since the response at the beginning of the pandemic in the U.S. varied across each state. Another strength of our study is that we conducted the online focus group discussions in the same manner as we would conduct them in a face-to-face setting. One of the benefits we observed of using Zoom for our focus group discussions was that we were able to have online focus groups that imitated the experience of face-to-face focus groups. Additionally, there were no deviations from our research protocol, and we were able to ensure the quality of the data. Lastly, we contracted an independent interviewer to conduct the focus group discussions and initiate the analysis to prevent bias.

## 5. Conclusions

The COVID-19 pandemic exacerbated health disparities that have been previously observed in Hispanics/Latinos. We observed how social capital served as an essential resource for Hispanics/Latinos in helping them navigate through the challenges of the coronavirus pandemic. Most importantly, we observed how social capital permeated the larger socioeconomic challenges faced over the course of the pandemic. While this social phenomenon was mostly positive for the Hispanic/Latino community, it also became a network through which misinformation could spread; this has troubling implications for vaccination rates in the population. This study points toward a larger discussion about the utility of social capital in guiding pandemic responses and recovery, particularly in Hispanics/Latinos in the U.S. Public health interventions that use social capital to address health disparities in marginalized communities are needed, especially in the context of emergency response events such as a pandemic. Additionally, more inquiry is necessary to examine how to increase trust and stop the spread of misinformation in U.S. Hispanics/Latinos.

## Figures and Tables

**Table 1 ijerph-20-05465-t001:** Demographic characteristics of focus group participants (*n* = 25).

Characteristic	Participants (%)
Age	
18–40	11 (43.6)
41–60	10 (39.1)
>60	4 (17.3)
Sex	
Male	6 (23.8)
Female	19 (76.2)
Income	
Less than $30,000	11 (45.5)
$30,000–$39,999	3 (13.6)
$40,000–$49,999	5 (18.2)
$50,000–$59,999	1 (4.5)
More than $100,000	5 (18.2)
Education	
Less than high school	2 (9.1)
Some high school	5 (18.2)
High school diploma	3 (13.6)
Some college	1 (4.6)
College	14 (54.6)
Primary language	
English	7 (28.6)
Spanish	18 (71.4)
Insurance status	
Yes	17 (66.7)
No	8 (33.3)
Employment type	
Essential worker	11 (42.9)
Non-essential worker	11 (42.9)
Not ascertained	3 (14.3)

**Table 2 ijerph-20-05465-t002:** Topics under investigation with themes and quotes derived from focus group discussions (n = 25).

Theme	Subtheme(s)	Translated Quote
Bonding social capital	Solidarity	Something that I noticed more of was solidarity. For instance, I now know almost all my neighbors in the building. There was a moment when the building didn’t have water, and in response, my neighbors gathered to have a meeting, which is something that never happened before. (Male, Hospital Worker)
	Solidarity	I think that people have gained more support from friends or neighbors. Because my wife and I were infected with COVID, and honestly our neighbors were very supportive. They brought us something to eat, or sometimes tea to drink. I think that I gained a little more trust with my neighbors. (Male, occupation not ascertained)
	Virtual meetings	I think that family unity is important, because even if it were by phone or by a FaceTime call, families remained connected. They would ask me for a session to be held so that various members who were in different homes could see the patient. In our culture, there is a union and a family love that is different from other cultures. (Female, Hospital Worker)
	Isolation	The beginning of the pandemic was very hard for me, because I was living in New York completely alone and I had no family nearby. Before, I had many friends and co-workers that helped me a lot when I first moved to New York. But with the pandemic, everyone was separated. As a result, I could not see my friends, and the interaction I had with my colleagues was simply over Zoom. The first few months of the pandemic were hard, but after I went to visit family, I began to feel better. (Female, Research Coordinator)
	Isolation	I socialized with an LGTBQ group from a Catholic parish, which is one of the few in Manhattan, but it has been very difficult for me to socialize with them during the pandemic because we only socialized through social media and videocalls, which does not compensate for hugs. It also doesn’t compensate for close conversations, or having the person in front of you, for you to see how that person really is doing. (Male, Court Translator)
Bridging social capital	Information exchange	As a medical professional, my job allows me to help and guide people, but I can’t prescribe or administer medications because of limitations from my license. However, I can focus on the preventive part of medicine, which includes demolishing cultural myths that prevent people from getting vaccinated right now. (Male, Internal Medicine Doctor, Community Health Organization Worker)
	Information exchange	My mom receives services from a retirement home and most of those workers are Latinas, from the Caribbean. Initially, when I asked them, “what information was the company giving you regarding COVID, or regarding the vaccine?” They told me that they were not receiving information and their personal notions were out of fear. I had to take the initiative to try to refer them to places in Spanish that informed them about COVID, and gave them information verbally, and gradually they changed their minds about the vaccine. Now one of them is fully vaccinated and the other one I helped her get a vaccinated. (Female, Semi-Retired Social Worker)
	Information exchange	One of the very big doctors we have at Mount Sinai taught a lecture about everything that had to do with the vaccine. The lecture covered the materials and the proteins they used in the vaccine. We were informed in detail. Especially for many of the people who are already older, who do not want to get the vaccine, he explained it very well and in detail everything about the vaccine, and how you are going to help each one, where to go. He gave us a lot of information. (Female, Hospital Spiritual Care Volunteer)
	Information exchange	We call the person, and we ask if they’ve been vaccinated, and if not, we complete the registration…that is why we go above and beyond, to improve our access to them…this is a direct action that we can do and that can be multiplied. In fact, I think that this program as a health promoter, has been for them a solution to many of their problems, or at least try to solve these problems and try to show them that it can be accessed, that with the necessary tools that we can provide, they can reach the systems, since the system is difficult. (Male, Internal Medicine Doctor, Community Health Organization Worker)
Trust	Vaccine distrust	I have decided to hold off getting vaccinated because I have seen cases of people who did not have the virus and then got it by getting vaccinated, or because I also heard in the news that certain people that got vaccinated later died two hours later. This creates a bit of fear because you do not know what reactions you are going to have in your own body. (Female, Essential Daycare Worker)
	Vaccine distrust	The only objection I have about the vaccine is the short amount of time it was developed. All medications have some side effect, but in the case of the vaccine we still don’t know much, and it may take up to 10 years for us to know. That is the only opposition that I have regarding the vaccine. (Female, Homemaker)
	Mistrust in politicians; mistrust in scientists	I don’t trust any politician. I feel like a lot of things are being done just to ease people’s fear, including vaccinations. They are rushing vaccinations without having long trials to see if it really is good. I don’t trust the scientists, or the government, in how they are handling the pandemic. (Female, Essential Daycare Worker)
	Mistrust in medicine	There is a lot of mistrust among the Latino population with medical services. I remember that many patients have told me of individuals that they know that go to the hospital that arrived alive and have not left. (Female, Essential Hospital Worker)
	Trust in the vaccine	I’m not afraid of vaccines. It has been almost 30 years that I do not know what a cold is, because I have a habit between October and November to get vaccinated for influenza. I know they are efficient and that’s why I get vaccinated with everything they offer me in my medical plan. (Male, Retired, Part-Time Construction Worker)
	Trust in politicians; trust in scientists	The difference with this crisis is that this is a health crisis where a politician must share the leadership space with the scientific and medical community. Right now, Dr. Fauci, even if he doesn’t want to, he has become an example, because he is the expert, and he must come out and show his face. He explains to the public through social media or television or whatever. The politician will always be a politician with or without a virus. I believe that in this case when the stage or scene is shared with the scientist, the situation is already different, and I am confident. (Male, Court Translator)
	Faith	At first, I said that I was not going to get vaccinated, because some people said that it was going to change your DNA. I don’t like listening to people, and I always believe a lot in my God. (Female, Hospital Spiritual Care Volunteer)
Dark side of social capital	Caregiving burden	When I was in New York, I spent most of the pandemic there. I was always calling my grandmother, checking-in to see that she was fine. As [the pandemic] was going on, I began to worry much more about my grandmother. Mostly about her being alone. I decided to come and stay with her, since I am still teleworking. At that time, I stopped worrying about myself, but worried more about her well-being. I didn’t want her to get out of the house and get infected. Now, I feel better being here with her. I feel more at peace and happy that she already has someone, and that she is not alone. (Female, Research Coordinator)
	Caregiving burden	Now that we are all in the house, it has not been easy. My mother obviously with dementia has many needs and a lot of confusion. Although she has services, they are partial, so I frequently must talk to her, and reassure her everything will be okay. (Female, Semi-Retired Social Worker)
	Misinformation	As much as one always tries to convince Hispanic people to go to the doctor so that they can take better care of themselves, they just don’t go, and they mostly rely on home remedies. For example, a friend of mine told me, “Don’t get the vaccine. I’ll bring you some pills from Santo Domingo instead that won’t give you COVID.” However, I don’t believe in any of that. (Female, Hospital Spiritual Care Volunteer)
Emergent theme: structural racism	Lack of resources	Some Latinos do not have safety-net resources or social security that the rest of the population that has legal status have. Most of our population do not have a legal status in this country and do not have access to many things. (Female, Hospital Worker)
	Lack of resources	We have seen that the systems have been designed to have access to vaccine information or appointments, for many members of our community, particularly the elderly, but it requires a computer, which they do not have access to, or do not have access to the Internet. If people are working, they must make a phone call, but the wait is so long that they can’t be on the phone waiting for an appointment. (Female, Semi-Retired Social Worker)
	Migratory status	I work with people who are undocumented, and if they feel sick, they don’t have the money to pay for medical consultations and they’re afraid to go to the clinics. I have worked with many clients who have had COVID, and they are very vulnerable because they either don’t have the academic knowledge or have the education, and they do not speak English well. They also do not have an insurance plan or if they do, they are people who need so much, so they go to work even when they are sick. (Male, Court Translator)
	Health insurance	Access to health services and health insurance is obviously a very serious problem in our community. All people who work and do not have medical insurance, they often do not seek medical services. The issue of access is also that the medicines are very expensive, and even though you have insurance, you must pay a co-pay or whatever. Sometimes you must make the decision of either paying the rent and the food for the month or pay for the medicines. (Female, Semi-Retired Social Worker)

## Data Availability

Transcriptions of the interviews are available upon request.

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
