# Peer review of "Emergent Social Capital during the Coronavirus Pandemic in the United States in Hispanics/Latinos"

_ijerph, 2023, doi:10.3390/ijerph20085465_

Round 1

Reviewer 1 Report

Dear authors

Thank you for giving me the opportunity to read your paper. I have a few suggestions that you may consider as you develop the paper further:
Positioning, purpose, introduction and research gap.

• The introduction should clearly illustrate (1) what we know (the key theoretical perspectives and empirical findings) and what do we not know (major, unaddressed puzzle, controversy, or paradox does the study addresses, or why it needs to be addressed and why this matters). And, (2) what will we learn from the study and how does the study fundamentally change, challenge, or advance scholars’ understanding. Much sharper problematization is required so that the introduction draws the reader into the paper. The introduction therefore needs to do a better job in setting the stage for the articulation of the theoretical contributions of the study. At the end of the introduction, we should have a clear idea of what the paper is about (i.e. its motivation, the gap in understanding that the paper is trying to address and summary of theoretical contributions).
Literature review
The paper should relate coherently and convincingly with issues of real-world significance. This is a crucial phase contributing to research design.
Suggestions
• Add more information to enable readers' understanding of the authors' view.
•           The article is in need of professional English proofreading in order to amend the various language shortcomings.
Method
•           Insufficient transparency. The authors need to provide much more details on their method, including data gathering and data analysis.
•           Why did you choose this analyse for your research?
Findings and discussion
Needs clear and comprehensive explanations to assist readers' understanding.
Conclusion
The conclusion falls short of providing sufficient information that would allow a reader to understand the contribution of this research.  What was found?
How does your review help other authors researching this?

Using the following references could be beneficial as these add more evidence to the literature review section:

 (2021). Investigating social capital, trust and commitment in family business: Case of media firms. Journal of Family Business Management.  (2019). The effect of human and social capital on entrepreneurial activities: A case study of Iran and implications. Entrepreneurship and Sustainability Issues, 6(3).

Best of luck with the further development of the paper.

Author Response

Thank Reviewer 1 for taking the time to review our manuscript and provide us with helpful comments to improve the draft. Please see the attachment with the responses to your comments.

Reviewer 2 Report

In the abstract should be added suggestions or recommendations for further research.

This research is interesting, however, the sample data is very small. We recommend that the sample data can be added with a cross section or times series min 30, especially the addition of Male. So that if the sample data is more, it will be closer to the actual data.

In Table 2. Topics under investigation with themes and quotes derived from the interviews (n=25). Need to provide a reference source. So it's more informative

In this background, it needs to be strengthened by the latest phenomena. In articles, especially literature reviews, it is better to strengthen it with grand theory.

Author Response

Thank Reviewer 2 for taking the time to review our manuscript and provide us with insightful comments to improve the draft. Please see the attachment with the responses to your comments.

Reviewer 3 Report

Thank you for the opportunity to review this manuscript. I have some comments:

1- In the methodology section, please add subtitle : participants (sampling).

2- In the methodology section please add subtitle : instrument and must be explained how the instrument was developed.

3- In the methodology section please add subtitle :validity and reliability.

4- In the discussion section please add compare between the results of previews studies and the results of current studies.

5- please cite this article :

Al-Tit, A.A.; Al-Ayed, S.; Alhammadi, A.; Hunitie, M.; Alsarayreh, A.; Albassam, W. The Impact of Employee Development Practices on Human Capital and Social Capital: The Mediating Contribution of Knowledge Management. J. Open Innov. Technol. Mark. Complex. 2022, 8, 218. https://doi.org/10.3390/joitmc8040218

Author Response

Thank Reviewer 3 for taking the time to review our manuscript and provide us with insightful comments to improve the draft. Please see the attachment with the responses to your comments.

Reviewer 4 Report

Dear authors,

thank you so much for your efforts in scientific research. This article makes a valuable contribution to understanding the impact of the COVID-19 pandemic on marginalized communities in the United States, particularly Hispanics/Latinos. The use of qualitative interviews and focus groups via Zoom provides rich insights into the role of bonding social capital, linkage social capital, and trust in this community during the pandemic. The findings highlight the importance of social capital in addressing the socioeconomic challenges of Hispanics/Latinos and the negative consequences that can result, such as caregiver burden and misinformation. The study's recommendation to invest in social networks and social capital for public health interventions is particularly timely and important given the vulnerability of sociodemographic groups during a public health crisis. Overall, this is an important study that has the potential to inform policies and interventions to support marginalized communities.

Thanks again for your contribution.

Best regards.

Author Response

Thank Reviewer 4 for taking the time to review our manuscript and provide us with encouraging comments on our study. Please see the attachment with the response to your comment.

Reviewer 5 Report

Dear Authors,

The paper entitled “Emergent Social Capital during the Coronavirus Pandemic in the United States in Hispanics / Latinos” describes how bonding social capital, bridging social capital, and trust played a role in Hispanics/Latinos over the course of the COVID-19 outbreak. Furthermore, the authors explore the negative consequences of social capital.

The paper's topic suits the aim and scope of the journal “International Journal of Environmental Research and Public Health,” and is interesting and relevant. All the cited references are relevant to the research. The paper has a sufficient level of novelty and practice recommendations. Overall, the article can be interesting for potential readers. 

The presented methodology, results, discussion, and conclusion are adequate. However, N=25 is not sufficient for the reliability of the received results. Despite my scientific doubt, the value of the article remains the same.

I would like to draw authors’ attention to some points in the article to improve the value of their issue:

Please add a short description of the paper at the end of the section “Introduction.”

Please add a paragraph at the end of the section “conclusion” related to the perspective of further research on the basis of the presented research.

I can recommend this paper for publication in International Journal of Environmental Research and Public Health after minor revision. 

Best wishes,

The reviewer

Author Response

Thank Reviewer 5 for taking the time to review our manuscript and provide us with insightful comments to improve the draft. Please see the attachment with the responses to your comments.

Round 2

Reviewer 1 Report

Dear authors

Hope you are doing well. According to the review of this article, the corrections have been made.

Good luck